**Editor:** Álvaro Acosta-Serrano, University of Notre Dame, UNITED STATES OF AMERICA

**Data Availability Statement:** All datasets that are not included in the manuscript are available in the

# Branched-chain amino acids modulate the proteomic profile of *Trypanosoma cruzi* metacyclogenesis induced by proline

**Janaina de Freitas Nascimento**[1]☯*, **Flávia Silva Damasceno**[1]☯, **Sabrina Marsiccobetre**[1]☯, **Francisca Natália de Luna Vitorino**[2], **Renan Weege Achjian**[1], **Julia Pinheiro Chagas da Cunha**[2], **Ariel Mariano Silber**[1]*

1 Laboratory of Biochemistry of Tryps - LaBTryps - Department of Parasitology, Instituto de Ciências Biomédicas II, Universidade de São Paulo, São Paulo, Brazil, 2 Laboratório de Ciclo Celular - Instituto Butantan, São Paulo-SP, Brazil; Centro de Toxinas, Resposta Imune e Sinalização Celular (CeTICS), Instituto Butantan, São Paulo, Brazil

☯ These authors contributed equally to this work.
* janainafn@usp.br (JFN); asilber@usp.br (AMS)

## Abstract

*Trypanosoma cruzi*, the causative agent of Chagas disease, has a complex life cycle that involves triatomine insects as vectors and mammals as hosts. The differentiation of epimastigote forms into metacyclic trypomastigotes within the insect vector is crucial for the parasite's life cycle progression. Factors influencing this process, including temperature, pH, and nutritional stress, along with specific metabolite availability, play a pivotal role. Amino acids like proline, histidine, and glutamine support cell differentiation, while branched-chain amino acids (BCAAs) inhibit it. Interestingly, combining the pro-metacyclogenic amino acid proline with one of the anti-metacyclogenic BCAAs results in viable metacyclics with significantly reduced infectivity. To explore the characteristics of metacyclic parasites differentiated in the presence of BCAAs, proteomics analyses were conducted. Metacyclics obtained in triatomine artificial urine (TAU) supplemented with proline alone and in combination with leucine, isoleucine, or valine were compared. The analyses revealed differential regulation of 40 proteins in TAU-Pro-Leu, 131 in TAU-Pro-Ile, and 179 in TAU-Pro-Val, as compared to metacyclics from TAU-Pro. Among these, 22%, 11%, and 13% of the proteins were associated with metabolic processes, respectively. Notably, enzymes related to glycolysis and the tricarboxylic acid (TCA) cycle were reduced in metacyclics with Pro-BCAAs, while enzymes involved in amino acid and purine metabolic pathways were increased. Furthermore, metacyclics with Pro-Ile and Pro-Val exhibited elevated enzymes linked to lipid and redox metabolism. The results revealed five proteins that were increased and four that were decreased in common in the presence of Pro+BCAAs, indicating their possible participation in key processes related to metacyclogenesis. These findings suggest that the presence of BCAAs can reshape the metabolism of metacyclics, contributing to the observed reduction in infectivity in these parasites.

public repository ProteomeXchange Consortium via the PRIDE partner repository (PXD049190).

**Funding:** This work was supported by: Fundação de Amparo à Pesquisa do Estado de São Paulo (FAPESP) 2021/12938-0 (awarded to AMS), and 2018/15553-9 (awarded to JCPC) Conselho Nacional de Pesquisas Científicas e Tecnológicas (CNPq) grant 307487/2021-0 (awarded to AMS) and Wellcome Trust grant 222986/Z/21/Z (awarded to JFN and AMS). The funders had no role in study design, data collection and analysis, decision to publish, or preparation of the manuscript. JFN is a Wellcome Trust fellow. FSD, SM and FNLV were FAPESP fellows during the development of this work.

**Competing interests:** The authors have declared that no competing interests exist.

## Author summary

*Trypanosoma cruzi*, the parasite causing Chagas disease, infects 6–7 million people in the Americas. Its life cycle involves the infection of triatomine insects which transmit the infection to humans and other mammals. Inside the insects, the parasite proliferates as non-infective forms, and its transmission to mammals depends on their differentiation into infective forms. Proline is a factor supporting this differentiation, while valine, leucine and isoleucine (known collectively as BCAA for branched-chain amino acids) do not. Combining proline with each BCAA results in a diminished number of viable parasites, with significantly reduced infectivity. To explore the causes of such diminished infectivity, we performed a proteomics analysis to compare parasites obtained with proline, in the presence or absence of BCAA. We observed that BCAA differentially regulated different proteins associated with metabolic processes: BCAA diminished enzymes related to glycolysis and the tricarboxylic acid (TCA) cycle while increased enzymes involved in amino acid and purine metabolism. Furthermore, isoleucine or valine increased enzymes linked to lipid and redox metabolism. These findings suggest that the presence of BCAAs can reshape the metabolism of the parasite, contributing to the observed reduction in infectivity.

## Introduction

*Trypanosoma cruzi* is the causative agent of Chagas disease, a neglected disease endemic of the Americas and affecting worldwide 6–7 million people [1]. *T. cruzi* has a complex life cycle occurring among triatomine insect vectors and mammals. Inside the triatomines, the parasites proliferate as non-infective epimastigotes, which colonize the digestive tube. In the terminal portion of the digestive tube, epimastigotes differentiate into the infective, non-proliferating forms metacyclic trypomastigotes, which are responsible for initiating the infection in mammals, including humans. The differentiation process from epimastigotes to metacyclic trypomastigotes is known as metacyclogenesis. This process occurs naturally at the rectal ampoule of the triatomine [reviewed by [2]], and can be reproduced *in vitro* by using the chemically defined medium denominated triatomine artificial urine (TAU), which mimics the physico-chemical characteristics of the insect urine [3]. TAU medium must be supplemented with nutrients that are able to support metacyclogenesis after a starvation period [3,4]. The most commonly used formulations are the ones supplemented with a combination of glucose (Glc) and the amino acids proline (Pro), glutamate (Glu) and aspartate (Asp) (TAU 3AAG) [3]. However, it has been shown that some amino acids can support metacyclogenesis *in vitro* by themselves [4,5]. It is tempting to attribute the metacyclogenic activities of metabolites such as glucose, Pro, Asp, Glu and glutamine (Gln) to their demonstrated role as carbon and energy sources. However, not every amino acid fueling ATP production is able to support metacyclogenesis: epimastigotes do not differentiate in the presence of the branched-chain amino acids (BCAAs) leucine (Leu), isoleucine (Ile) and valine (Val) as the only carbon source in TAU [3,4]. Even more, when TAU-Pro, is supplemented with any of the BCAAs metacyclogenesis is significantly diminished [4], and very interestingly, the diminished population of metacyclics that can be obtained in the presence of BCAAs are less infective [6]. Thus, regardless of their role in ATP production, some amino acids can support metacyclogenesis, acting as 'pro-metacyclogenic' and others somehow inhibit this process, acting as 'anti-metacyclogenic'.

Proline can induce metacyclogenesis in different *T. cruzi* strains such as Dm 28, Dm 30, CL, Perú isolate, CL Brener and CL14 [3,4,6], and the participation in this process of each

enzyme involved in proline oxidation has been analyzed [7]. In addition to this, proline also plays an important role in many other biological processes in *T. cruzi*, such as energy metabolism and serving as a carbon source [8–10], intracellular differentiation [11], host-cell invasion [12], resistance to thermal and nutritional stress [13] oxidative imbalance [9,13,14] and regulatory volume decrease in response to osmotic stress [15]. In order to be available for the many functions it is involved in, proline is obtained by *T. cruzi* from the extracellular environment [16] and is able to biosynthesize it from glutamate [17,18]. The oxidation pathway of proline to glutamate occurs in two enzymatic and one non-enzymatic steps: first proline dehydrogenase (ProDH) converts proline into $\Delta^1$-pyrroline-5-carboxylate (P5C) [9]. P5C undergoes a non-enzymatical chemical reaction by hydrolyzing the pyrrolic ring, rendering glutamate-γ-semialdehyde (GSA), which is the substrate of the P5C dehydrogenase (P5CDH), which finally oxidizes it into Glu [8]. In yeast, P5CDH can be inhibited by leucine and isoleucine, [19] and in *T. cruzi* leucine at high concentrations also inhibits P5CDH [6].

Given that: (i) proline is a 'pro-metacyclogenic' amino acid; (ii) BCAAs do not support the differentiation from epimastigote to trypomastigote metacyclic; (iii) BCAAs impair metacyclogenesis supported by proline and (iv) metacyclics obtained in the presence of BCAAs are less infective, we analyzed the global proteomic profiles of the metacyclic trypomastigotes obtained from TAU medium supplemented with different metabolic substrates: 3AAG, Pro, Pro-Leu, Pro-Ile and Pro-Val. Globally, we found that parasites differentiated in TAU-3AAG and TAU-Pro medium are very similar, whereas the comparison between metacyclics from TAU-Pro versus TAU-Pro-BCAAs showed changes in the proteomic profile, mostly in proteins involved in metabolic processes, translation, proteolysis and regulation of gene expression.

## Methods

### *Trypanosoma* strains and culture

Epimastigotes from CL strain clone 14 were maintained in the exponential growth phase by subculturing every 48 h in LIT medium [20] supplemented with 10% FCS at 28˚C.

### Metacyclogenesis and metacyclic trypomastigotes purification

Stationary phase epimastigotes were obtained from a starting exponential cell culture at $5 \times 10^6$ parasites/mL, kept for 4 days in LIT at 28˚C. Parasites were washed with PBS (137 mM NaCl; 2.6 mM KCl; 8 mM $Na_2HPO_4$; 1.4 mM $KH_2PO_2$, pH 7.4) at 1,600 x *g* for 5 min, transferred to TAU (190 mM NaCl; 17 mM KCl; 2 mM $MgCl_2$; 2 mM $CaCl_2$; 8 mM potassium phosphate buffer, pH 6.0) at a final concentration of $5 \times 10^7$ parasites/mL and incubated for two hours at 28˚C. After, parasites were harvested and transferred to one of the TAU media of interest: TAU 3AAG (TAU + 10 mM Glc; 10 mM Asp; 10 mM Glu; 10 mM Pro; pH 6.0), TAU Pro (TAU + 10 mM Pro), TAU Pro-Leu (TAU Pro + 10 mM Leu), TAU Pro-Ile (TAU Pro + 10 mM Ile), TAU Pro-Val (TAU Pro + 10 mM Val) at a final concentration of $5 \times 10^7$ parasites/mL. After 6 days of incubation, the differentiation rate was estimated by counting the parasites in the Neubauer chamber. Then, parasites were washed twice with PBS, pH 8.0 and the metacyclic trypomastigotes forms were isolated by ion exchange chromatography on DEAE-cellulose (Sigma) as previously described [21].

### Protein extraction and digestion

Approximately $1 \times 10^8$ purified metacyclic trypomastigotes obtained in each differentiation condition were used to prepare protein extracts. Cells were resuspended in 250 μL lysis buffer (8 M urea, 75 mM NaCl, 50 mM Tris, pH 8.2) supplemented with protease / phosphatase /

deacetylase inhibitors: 2 mg/L leupeptin, 1 μM pepstatin and 10 mM pyrophosphate, 0.1 mM sodium orthovanadate, 0.1 mM PMSF, 1 mM glycerophosphate, 5 mM sodium butyrate, 1 mM NaF. After protein quantification, 60 μg of proteins were added to 100 mM Tris-HCl pH 8.5. Next, trichloroacetic acid (TCA) was added to a final concentration of 20%. Samples were incubated for 16 h at 4°C and centrifuged at 14,000 x *g* for 30 min at 4°C. Finally, pellets were washed twice with 500 μL of cold acetone (14,000 x *g* for 10 min) and kept at room temperature to dry. The pellets from precipitation were resuspended in 30 μL of lysis buffer (8 M urea, 75 mM NaCl, 50 mM Tris pH 8.2) and reduced by adding 5 mM Dithiothreitol (DTT) for 25 min at 56°C. Alkylation was performed by adding 14 mM iodoacetamide for 30 min at room temperature in the dark. Next, 5 mM of DTT was added, and samples were incubated for another 15 min at room temperature in the dark. The urea concentration was reduced with 25 mM Tris pH 8.2 and 100 mM $CaCl_2$. Finally, trypsin (1 μg/μL; Sigma-Aldrich) was added in the proportion of 1:200 (enzyme:substrate) for 16 h at 37°C under agitation. After incubation, the reaction was stopped with 1 μL of 100% trifluoroacetic acid (TFA) and dried in speed-vac. Digested peptides were desalinized using the Sep-Pak Light tC18 Cartridges (Waters, Milford, MS, USA) columns. First, the columns were conditioned with 100% methanol, followed by a solution of 0.1% TFA and 50% acetonitrile (ACN) and a solution of 0.1% TFA. Next, samples were resuspended with 0.1% TFA and loaded to the column. After washing the columns with 0.1% TFA, samples were eluted with 0.1% TFA/50% ACN solution, dried in the speed-vac, and analyzed by mass spectrometry.

## Mass spectrometry-based analysis of *T. cruzi* tryptic peptides

Digested peptides from three biological replicates were resuspended in 0.1% formic acid and fractionated on an in-house reverse-phase capillary column (10 cm × 75 μm, filled with 5 μm particle diameter C18 Aqua resins-Phenomenex) coupled to nano-HPLC (NanoLC; Thermo-Fisher Scientific). The fractionation was carried out through a 60 min gradient of 5–35% gradient ACN in 0.1% formic acid followed by 30 min gradient of 35–95% at a flow rate of 200 nl/min. The eluted peptides were analyzed online on the LTQ-Orbitrap Velos High-Resolution Mass Spectrometer (ThermoFisher Scientific) (source voltage of 1.9 kV, capillary temperature of 200°C). The mass spectrometer was operated in DDA mode with dynamic exclusion enabled (exclusion duration of 45 s), MS1 resolution of 30,000, and MS2 normalized collision energy of 35. For each cycle, one full MS1 scan range of 200–2,000 m/z was followed by ten MS2 scans (for the most intense ions) using an isolation window size of 2.0 m/z and collision-induced dissociation (CID).

## Analysis and integration of proteomic data

Data from LTQ Velos-Orbitrap were analyzed using the MaxQuant (v. 1.6.1) software using the TriTrypDB database (taxonomy DB32 T. *cruzi* CL Brener Esmeraldo like and Non-Esmeraldo-like haplotype). Parameters were set with tolerance in the MS of 4.5 ppm and MS/MS of 0.5 Da, carbamidomethylation of cysteine as a fixed modification: oxidation of methionine and acetylation of protein N-termini as variable modifications; and a 1% False Discovery Rate (FDR). The match between runs option was used to increase the number of trusted IDs. Normalized LFQ values of the proteingroups. txt output were used to obtain the quantitative values. Next, Perseus (v. 1.5.6.0 and 1.6.2.1) program (from the MaxQuant package) identified differentially expressed proteins and integrated the data with available public information. Those proteins identified as potential false positives (contaminants, reversals, and identified by only one site) were removed. Finally, statistical tests (T-test with FDR permutation, FDR: 0.05; s0: 0.1) were used to identify proteins differentially expressed between conditions, accepting

only reproducible results. As a result, the p-value may vary between pairwise comparisons [22]. The mass spectrometry proteomics data have been deposited to the ProteomeXchange Consortium via the PRIDE partner repository (PXD049190).

### Bioinformatic analysis

For cluster analysis by the Fuzzy c-Means (FCM) algorithm, average LFQ values for each protein of each analyzed condition were used to generate clusters using http://computproteomics. bmb.sdu.dk/Apps/FuzzyClust/ [23]. Gene Ontology Biological Process and Cell Component analysis were performed using the tools available on TriTrypDB (https://tritrypdb.org/ tritrypdb/) and data were manually curated. The frequency of appearance of a determined GO was normalized over the number of proteins of each group. Heatmaps and bar graphs were generated using the GraphPad Prism version 10.1.0. Venn diagrams were generated using InteractiVenn (https://www.interactivenn.net/) [24]. The PCA decomposition was performed using MATLAB's "pca" function with default parameters (that is, performing data centering, attributing the same weight to all data points and using singular value decomposition as the underlying algorithm).

### Western blotting

Western blotting was used to determine the expression levels of tyrosine aminotransferase (TAT) in total cell lysates of metacyclic trypomastigotes of two biological replicates. Briefly, parasites were harvested as described above and resuspended in lysis buffer (20 mM Tris-HCl pH 7.9; 0.25 M sucrose; 1 mM EDTA pH 8.0; 0.1% (v/v) Triton X-100; 1 mM PMSF; 2 μg/mL aprotinin; 0.1 mg/mL TLCK and 10 μM E-64). Samples were sonicated on ice and clarified by centrifugation (10,000 x $g$ for 30 min at 4°C). Protein concentration was determined by the Bradford method using bovine serum albumin (BSA) as a standard [25]. Equal amounts of protein samples (30 μg) were loaded per lane and submitted to electrophoresis (SDS-PAGE) in gels containing trichloroethanol (Sigma-Aldrich). Proteins were transferred into 0.2 μm nitrocellulose membranes (BioRad), blocked in PBS-0.1% Tween-20 (PBST) supplemented with 5% (w/v) skimmed milk powder and probed (1 h at RT) against specific sera. The enzyme TAT was probed with a polyclonal rabbit antiserum (1:500), gently gifted by Prof. Cristina Nowicki, University of Buenos Aires [26]. The fluorescence signal of the total protein for each extract contained in the membranes after UV light exposure was used as the loading control. Membranes were washed three times and incubated with anti-mouse IgG or anti-rabbit IgG HRP-linked antibodies (Cell Signaling Technology) both diluted in PBST (1:2,000). The chemiluminescence reaction was performed using SuperSignal West Pico Chemiluminescent ECL substrate (Thermo Scientific) following the manufacturer's instructions. Bands corresponding to the detection of TAT were quantified by densitometry using the software ImageLab version 5.0.

### Pyruvate kinase activity assay

The activity of pyruvate kinase (PK) was determined in total cell lysates of metacyclic trypomastigotes obtained as previously mentioned. Equal amounts of protein extracts of two biological replicates of metacyclic trypomastigotes obtained in TAU Pro or TAU Pro-BCAAs (50 μg) were used for each reaction. The reaction mixture contained: 78 mM potassium phosphate buffer pH 7.6; 1.16 mM phospho(enol)pyruvic acid (PEP) (Sigma-Aldrich); 0.22 mM NADH disodium salt trihydrate (Amresco); 13.6 mM MgSO$_4$; 3 mM ADP sodium salt (Sigma-Aldrich); 10 U L-lactic dehydrogenase (Sigma-Aldrich). The reactions were monitored by spectrophotometry using a Spectramax i3 plate reader (Molecular Devices), at 28°C, in 96

wells plates following the decrease in absorbance at 340 nm due to NADH oxidation in the coupled lactate dehydrogenase reaction. For each reading, the same reaction mixture without PEP was used as control/blank. The media and standard deviation of two biological replicates were represented in a graph using GraphPad Prism 10 software.

## Results

### Metacyclogenesis in the presence of different amino acids induces distinct protein level profiles

To investigate the protein level profiles of the metacyclic trypomastigotes differentiated in the presence of different metabolites, we induced metacyclogenesis in TAU supplemented with either 3AAG, Pro, Pro-Leu, Pro-Ile, and Pro-Val (S1 Fig). Metacyclic trypomastigotes were purified and their proteome profile was evaluated by LC-MS/MS. After initial analysis, we identified 1,457 proteins as present in all samples (S1 Table). Principal component analysis showed that the proteomic profile of the metacyclics obtained in TAU-3AAG and TAU-Pro cluster together, whilst the metacyclics differentiated in the presence of Pro-BCAAs produce distinct proteomic profiles (Fig 1). Fuzzy c-means clustering [23] of LFQ values grouped approximately 28% of the identified proteins in 7 distinct clusters, evidencing different trends of protein abundance in each particular differentiation condition (S2 Table). To look in more detail at the proteins present in each cluster, we analysed the Gene Ontology annotation of biological process and cell compartment for each protein (Fig 2). The abundance of the proteins in Clusters 1 and 4 changed in response to the presence of either Pro-Ile or Pro-Val during cell differentiation. Approximately 18% of the proteins more expressed in these conditions (Cluster 1) participate in metabolic processes, mostly in amino acids metabolism, including tyrosine aminotransferase (TAT) which is responsible for the transamination step of the BCAAs degradation pathway [27]. Accordingly, proteins in Cluster 1 appear to be, aside from cytoplasmic, localised in mitochondria, glycosomes and the contractile vacuole. In contrast, less expressed proteins in metacyclics differentiated in TAU Pro-Ile and TAU Pro-Val (Cluster 4) are involved in oxidation-reduction, other metabolic processes, such as carbohydrates and purine metabolic processes, and post-transcriptional regulation of gene expression (Fig 2A, S3 and S6 Tables). Proteins involved in oxidation-reduction processes also appear less expressed in metacyclics differentiated in the presence of Pro-Leu (Cluster 2). Surprisingly, there is a high incidence of retrotransposon hot spot proteins (RHSPs) and proteins localised as integral components of membranes in Cluster 2 when compared to other clusters. On the other hand, proteins involved in post-transcriptional regulation of gene expression as well as in translation and RNA processing are up-regulated in metacyclics differentiated in TAU Pro-Leu (Cluster 3) (Fig 2B, S4 and S5 Tables). Cluster 5 group proteins which abundance reduces in cells differentiated in TAU Pro-Val. These are mostly involved in translation, protein folding and post-transcriptional regulation of gene expression, and include ribosomal proteins, translation initiation factors (eIFs) and RNA-binding proteins (RBPs). Correspondingly, these proteins mostly localise to the cytoplasm, nucleus, axoneme and ribosomes. In turn, proteins involved in proteolysis and other metabolic processes such as carbohydrates and fatty acids metabolic processes are up-regulated in response to the presence of Pro-Val during metacyclogenesis and localise to the mitochondria, glycosomes and contractile vacuoles, apart from the cytoplasm (Fig 2C, S7 and S9 Tables). Interestingly, cluster 6 shows proteins with increased abundance in TAU Pro-Ile, which mostly take part in translation, oxidation-reduction and amino acid metabolic processes, but there is no correspondent mirroring cluster (Fig 2D, S8 Table). Together, the data suggest that, although the BCAAs similarly inhibit metacyclogenesis supported by proline, they induce different proteomic profiles on the resulting trypomastigote metacyclics.

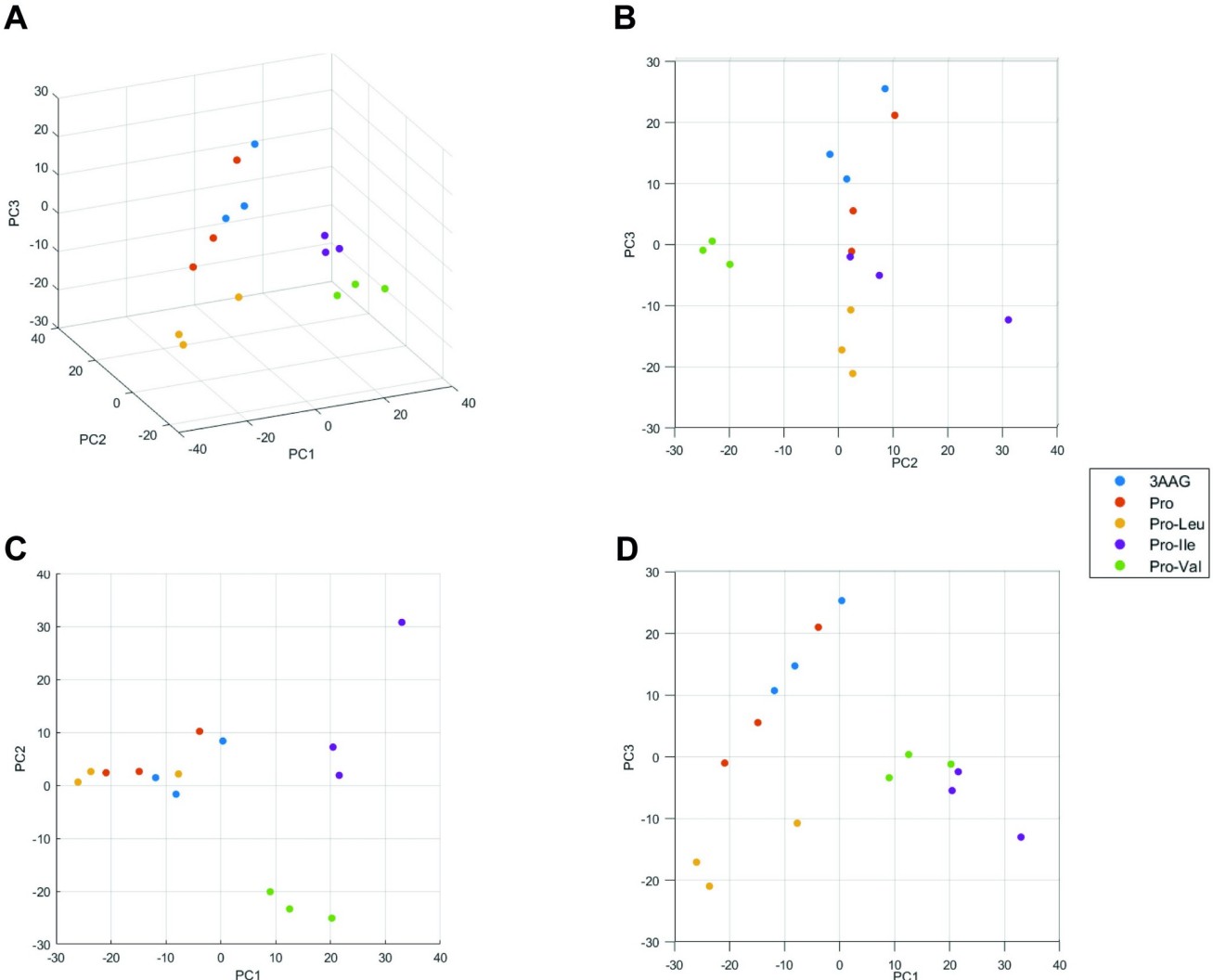

**Fig 1. Distribution of the experimental data in principal component space. A.** 3D view of the first three principal components, and projections on the planes defined by those principal components taken in pairs (**B.** PC3 x PC1; **C.** PC2 x PC1; **D.** PC3 x PC1). The variability explained by PC 1, 2 and 3 are, respectively, 27%, 16% and 14%. The other PCs explain 8% or less of the variability each. The points corresponding to 3AAG (blue) and Pro (purple) cluster together, with Pro-Leu (yellow) in its immediate vicinity, while Pro-Ile (orange) and Pro-Val (green) are further apart, both from the (3AAG, Pro, Pro-Leu) cluster and from each other. Each circle corresponds to an independent biological replicate for each differentiation condition.

Noteworthy, all clusters also point to a very similar trend of protein levels when we compare trypanosomes differentiated in TAU 3AAG and TAU Pro.

## Metacyclogenesis in TAU 3AAG and TAU Pro induce similar proteomic profiles

*In vitro* metacyclogenesis performed in TAU 3AAG and TAU Pro produce similar differentiation rates [28]. In order to evaluate whether metacyclic trypomastigotes differentiated in the presence of 3 amino acids plus glucose (TAU 3AAG) or just proline (TAU Pro) have also similar protein abundance profiles, we performed differential expression analysis. Metacyclic trypomastigotes obtained in TAU Pro presented a very similar proteomic profile when compared

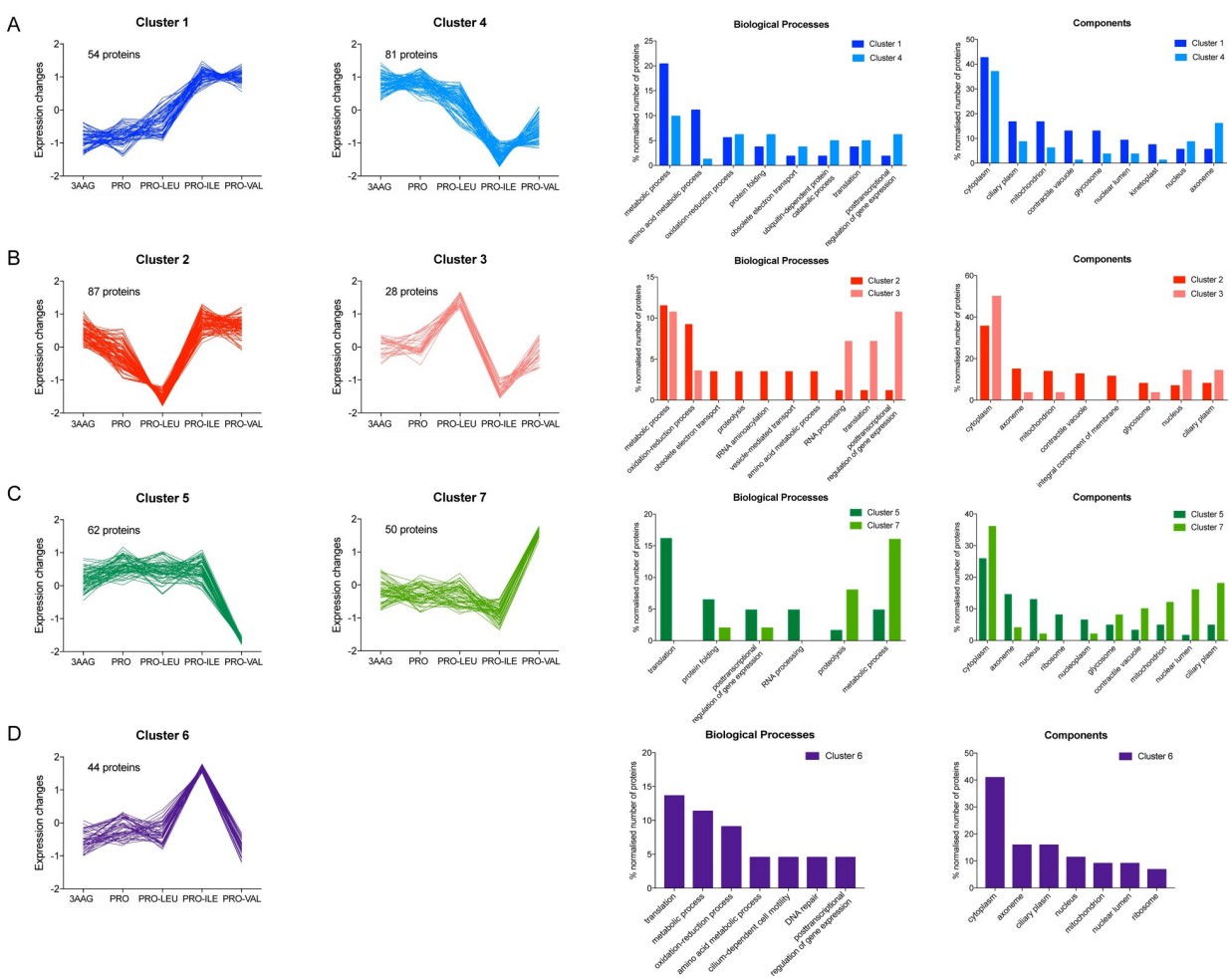

**Fig 2. Fuzzy c-means clustering analysis shows distinct protein level profiles in metacyclics differentiated in TAU 3AAG, Pro, Pro-Leu, Pro-Ile and Pro-Val.** Clusters evidence variations in the proteomic profiles of metacyclics differentiated in **A.** TAU Pro-Ile and Pro-Val; **B.** TAU Pro-Leu; **C.** TAU Pro-Val and **D.** TAU Pro-Ile. Grouped bar graph shows the distribution of the proteins of each cluster according to biological process and cell components. Average LFQ values were used for clustering analysis. Computed and Curated GO Biological Processes available in TriTrypsDB were used to generate bar graphs.

with metacyclics differentiated in TAU 3AAG. A few proteins are differentially expressed between these two differentiation conditions: 39 proteins in total, from which 17 are up-regulated and 22 are down-regulated in parasites differentiated in TAU Pro (S2 Fig, S10 and S11 Tables). Around 35% of the up-regulated proteins are annotated as hypothetical. Gene Ontology analysis evidence that most of these proteins are localized to cytoplasm, glycosome, nucleus, ribosome and axoneme. Up-regulated proteins are predominantly involved in energy production, nucleotide production, DNA organization and ribosome composition. On the other hand, of the 22 down-regulated proteins in TAU Pro parasites, only 18% are identified as hypothetical. Down-regulated proteins are predominantly related to the cytoplasm or nucleus and are mostly involved in energy metabolism and regulation of gene expression, respectively. Interestingly, proteins involved in metabolism such as proline racemase [29] and malate dehydrogenase [30] are downregulated in these parasites (S12 and S13 Tables).

### The relationship between BCAAs and Pro

Considering the importance of proline in metacyclogenesis and the negative interference of BCAAs, we compared the proteomic profile of parasites differentiated in TAU Pro medium with parasites differentiated in TAU Pro supplemented with each one of the BCCAs. We then excluded the proteins present in the differential expression analysis between metacyclic trypomastigotes from TAU 3AAG and TAU Pro because they are likely to be a result of the presence of other amino acids and glucose in the TAU 3AAG. Since different subsets of proteins are differentially expressed in response to the presence of each BCAA during metacyclogenesis, we describe the results of each comparison separately below.

### Comparison of proteomic patterns of metacyclic trypomastigotes from TAU Pro with TAU Pro-BCAAs

Leucine causes the strongest reduction in the metacyclogenesis induced by proline (see [4,6] and S1 Fig). However, differential expression analysis comparing the proteomic profiles of metacyclic trypomastigotes obtained in TAU Pro and TAU Pro-Leu shows only 47 differentially regulated proteins, from which 11 proteins were excluded for being present in the comparison between metacyclics differentiated in TAU 3AAG and TAU Pro. From the remaining proteins, 20 are down-regulated and 16 are up-regulated in parasites differentiated in TAU Pro-Leu (Fig 3A, S14 and S15 Tables). Approximately 35% and 25% of the proteins are annotated as hypothetical among the down and up-regulated datasets, respectively. In order to identify the biological processes in which the differentially expressed proteins are possibly involved, Gene Ontology analysis was performed (Fig 3B and 3C, S16 and S17 Tables). The results evidenced that down-regulated proteins are involved in biological processes such as the generation of energy, posttranscriptional regulation of gene expression and oxidation-reduction processes; on the other hand, up-regulated proteins participate in respiration and translation processes. In terms of cell components, down-regulated proteins are enriched in the cytoplasm and mitochondria when compared to the up-regulated proteins that are mostly present in axoneme, contractile vacuole and ribosomes.

The next step was to analyze the proteome changes induced in metacyclic trypomastigotes by the addition of Ile to the TAU Pro-defined medium. Differential protein abundance analysis comparing the proteomic profile of metacyclics obtained in TAU Pro and TAU Pro-Ile showed 154 differentially regulated proteins, from which 23 proteins were excluded for being present in the comparison between metacyclics differentiated in TAU 3AAG and TAU Pro. The remaining 131 proteins are divided into 68 down-regulated and 63 up-regulated proteins in metacyclics differentiated in TAU Pro-Ile when compared to the ones obtained in TAU Pro. Among the differentially expressed proteins, approximately 35% of the down-regulated proteins are annotated as hypothetical, whilst only 20% of the up-regulated proteins have the same annotation (Fig 4A, S18 and S19 Tables). Gene Ontology analysis showed that proteins putatively involved in translation, post-transcriptional regulation of gene expression and protein folding are down-regulated in response to the presence of Pro-Ile during cell differentiation, whereas proteins involved in ubiquitin-dependent protein catabolic process and amino acid metabolic processes are up-regulated. However, there are no major differences in the distribution of up and down-regulated proteins in the cell compartments (Fig 4B and 4C, S20 and S21 Tables). Noteworthy, there are possible changes in the BCAAs degradation pathway in parasites differentiated in TAU Pro-Ile, with up-regulation of tyrosine aminotransferase (TAT) [27] and down-regulation of the beta subunit of the branched-chain α-ketoacid dehydrogenase (BCKDH) complex, which performs the second step of the pathway [31].

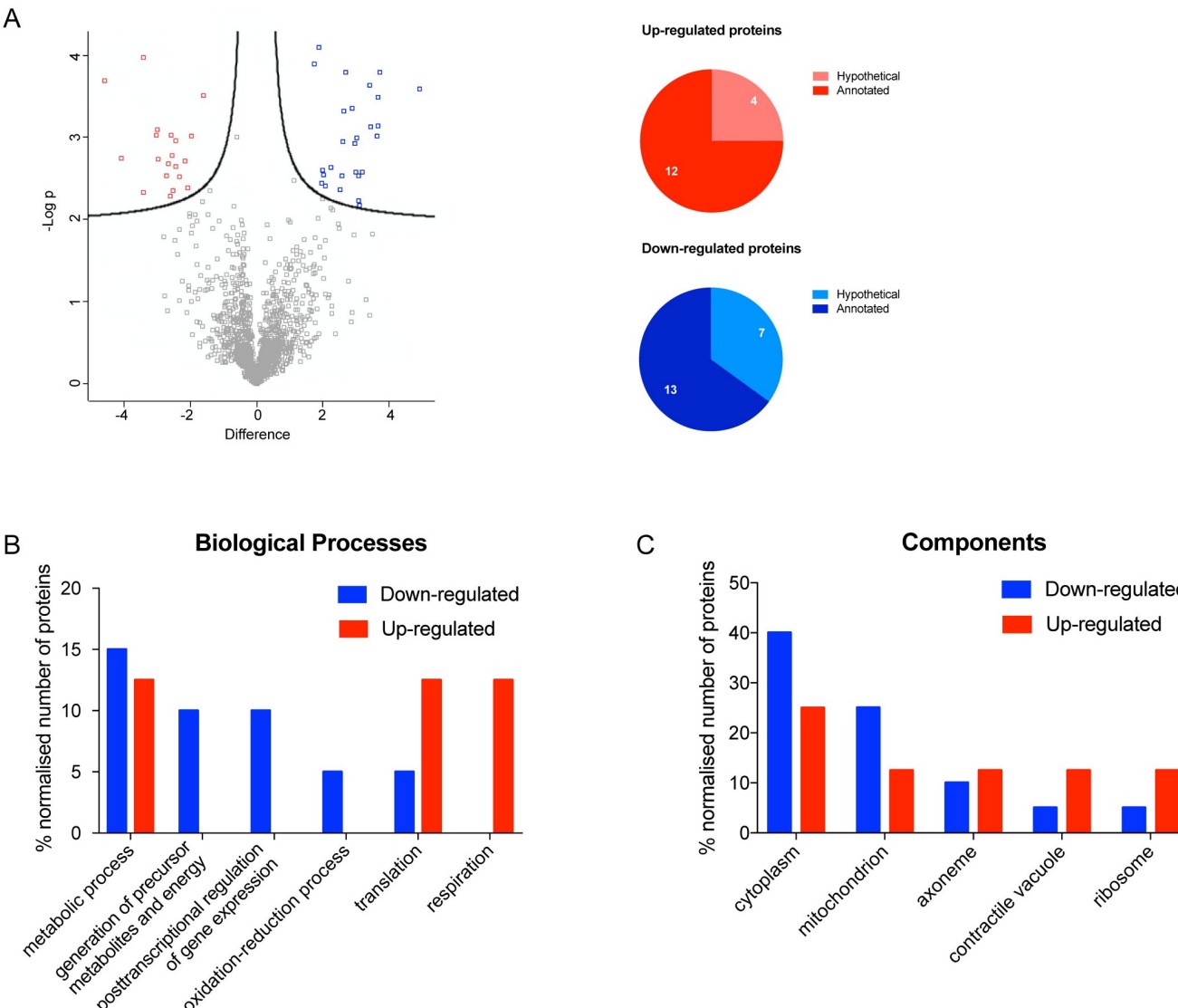

**Fig 3. Parasites differentiated in TAU Pro-Leu have a similar proteomic profile to those differentiated in TAU Pro. A.** Volcano plot of LFQ values of proteins from metacyclic trypomastigotes differentiated in TAU Pro and TAU Pro-Leu. Differentially expressed proteins are shown in blue (down-regulated in TAU Pro-Leu) and red (up-regulated in TAU Pro-Leu). Pie charts show the number of hypothetical proteins in the sets of up and down-regulated proteins. **B** and **C.** Distribution of up and down-regulated proteins according to biological process and cell component, respectively.

In contrast with the comparisons previously described, differential expression analysis between metacyclic trypomastigotes obtained in TAU Pro and TAU Pro-Val showed the highest number of differentially regulated proteins, 197 in total, from which 18 proteins were excluded for being present in the comparison between metacyclics differentiated in TAU 3AAG and TAU Pro. Among the remaining proteins, 78 are down-regulated and 101 are up-regulated in metacyclics obtained in TAU Pro-Val when compared to the ones differentiated in TAU Pro (Fig 5A, S22 and S23 Tables). Approximately 30% of the down-regulated proteins are annotated as hypothetical whereas only 20% of the up-regulated proteins have the same annotation. Gene Ontology analysis showed down-regulation of proteins involved in biosynthetic processes (such as translation and protein folding) and up-regulation of proteins

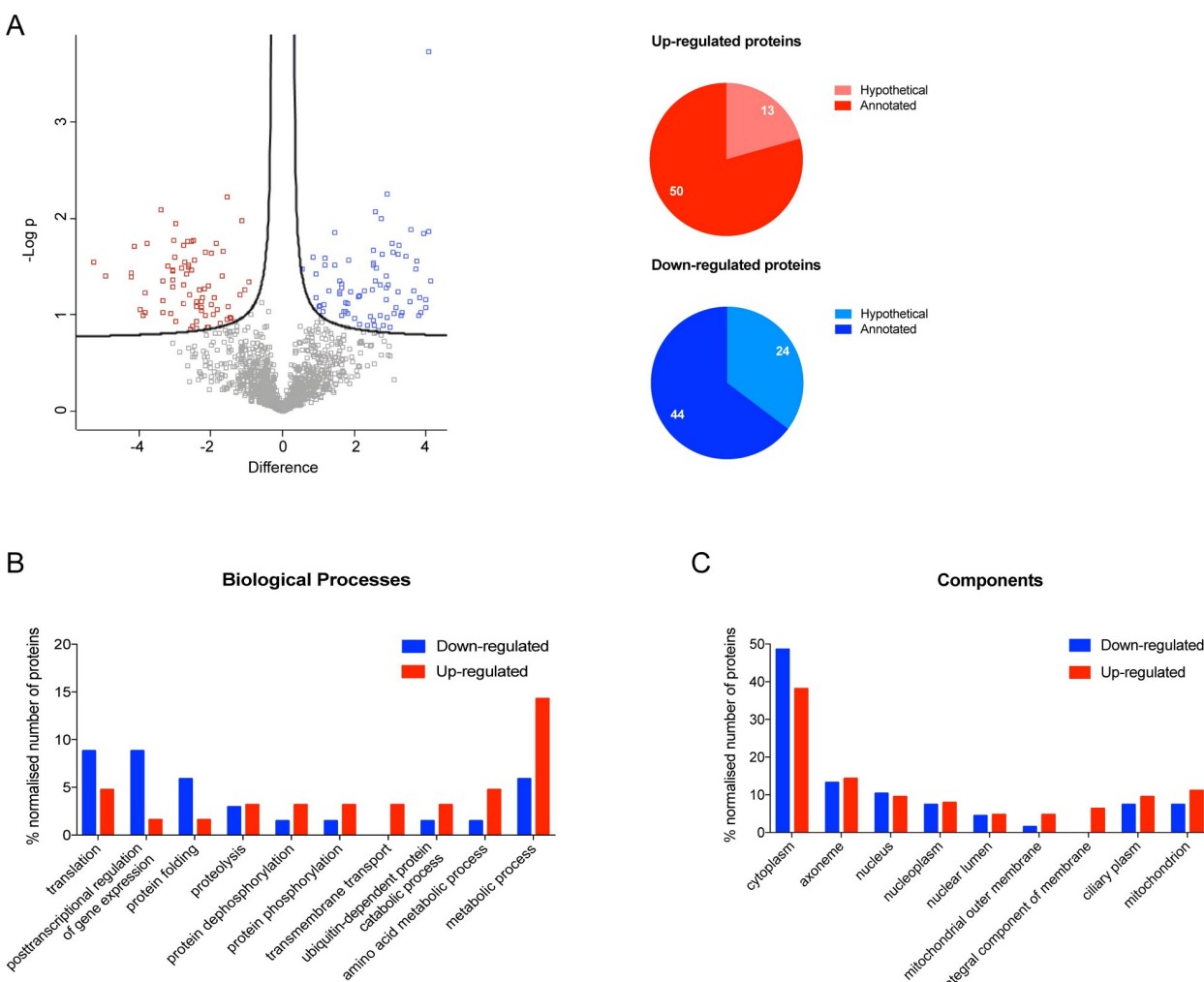

**Fig 4. Addition of Ile to TAU Pro induces proteomic changes in the resulting metacyclics. A.** Volcano plot of LFQ values of proteins from metacyclic trypomastigotes differentiated in TAU Pro and TAU Pro-Leu. Differentially expressed proteins are shown in blue (down-regulated in TAU Pro-Ile) and red (up-regulated in TAU Pro-Ile). Pie charts show the number of hypothetical proteins in the sets of up and down-regulated proteins. **B** and **C.** Distribution of up and down-regulated proteins according to biological process and cell component, respectively.

involved in metabolic processes (such as oxidation-reduction processes and proteolysis) (Fig 5B, S24 and S25 Tables).

Related to metabolic processes, results suggest that there are changes in the metabolism glutamate, proline, serine, threonine and glutamine. Unlike the previous conditions, there are changes in glucose metabolism. The enzymes phosphoglycerate kinase (PGK) [32], 2,3-bisphosphoglycerate-independent phosphoglycerate mutase (iPGAM), glycerol kinase (GK) and fructose-1,6-bisphosphatase (FBPase) that are involved in glycolysis and/or gluconeogenesis are differentially regulated. There is also differential regulation of two putative subunits of the respiratory chain–down-regulation of ubiquinol-cytochrome c reductase (UCYT CR) and up-regulation of cytochrome c oxidase VIII (COXVIII), also the putative ATP synthase is down-regulated, which might lead to alterations in energy production in these parasites. In the BCAAs degradation pathway, there is up-regulation of TAT, responsible for the first step of the pathway. Interestingly, proteins related to fatty acid metabolism are up-regulated, such as

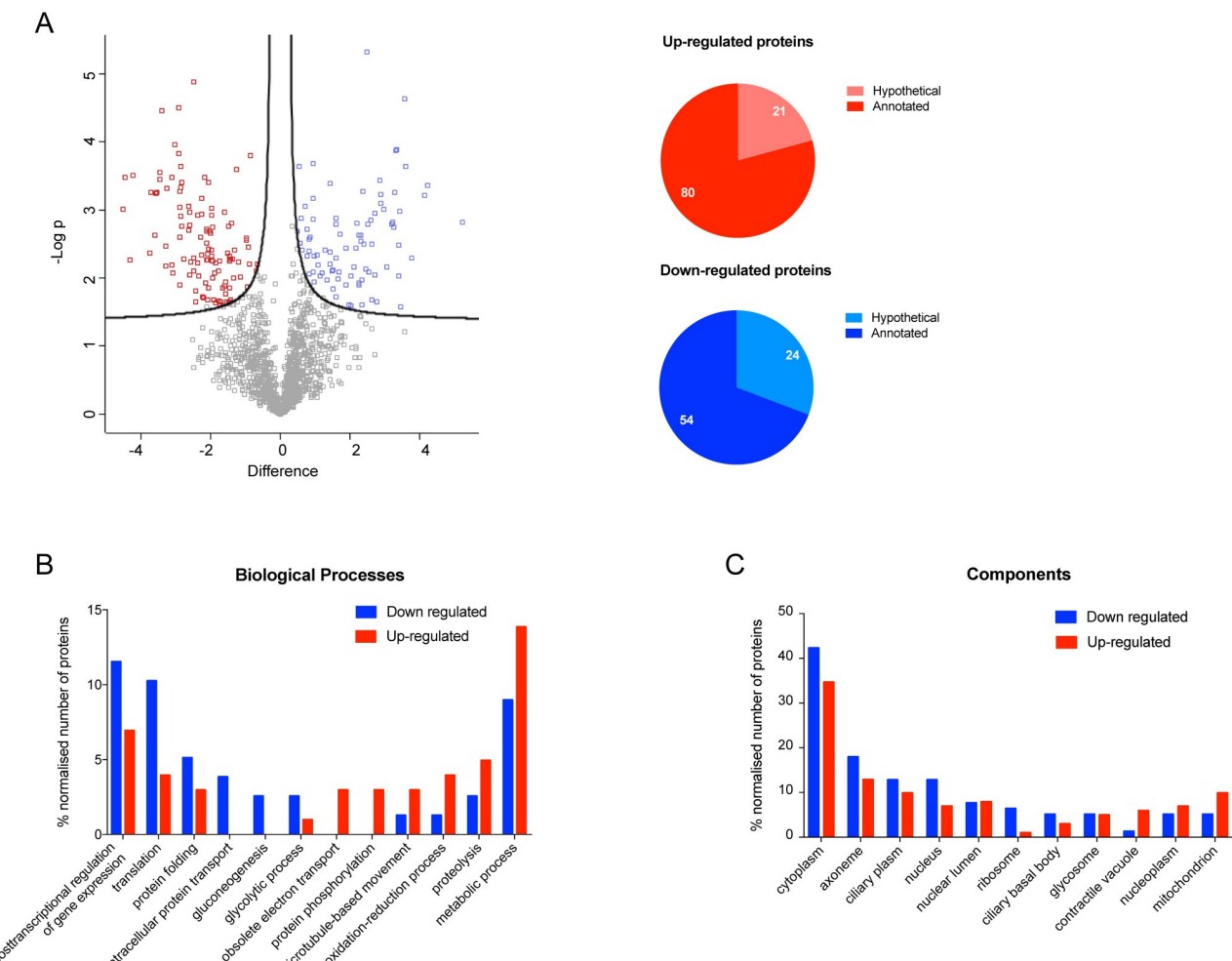

**Fig 5. Metacyclogenesis in TAU Pro-Val results in changes in proteins related to metabolic processes. A.** Volcano plot of LFQ values of proteins from metacyclic trypomastigotes differentiated in TAU Pro and TAU Pro-Leu. Differentially expressed proteins are shown in blue (down-regulated in TAU Pro-Val) and red (up-regulated in TAU Pro-Val). Pie charts show the number of hypothetical proteins in the sets of up and down-regulated proteins. **B** and **C.** Distribution of up and down-regulated proteins according to biological process and cell component, respectively.

fatty acyl-CoA synthetase 2 and carnitine O-palmitoyltransferase II, both putative. Moreover, proteins associated with the structure and motility of the cell are differentially expressed in metacyclics differentiated in TAU Pro-Val, such as the putative microtubule-associated protein Gb4 and flagellum attachment zone 4 which are down-regulated and the putative cytoskeleton-associated protein, flagellar member 1 and a trypanosoma basal body component are up-regulated.

## Final comparison TAU Pro x TAU Pro-BCAAs

Protein levels of over 300 proteins varied in response to the presence of proline combined with the different BCAAs during metacyclogenesis. These differentially regulated proteins are mostly involved in metabolic processes, posttranscriptional regulation of gene expression and translation (S3 Fig). However, only a few proteins are similarly differentially regulated when we compare the datasets of up and down-regulated proteins of metacyclic trypomastigotes

**Table 1. Proteins commonly differentially expressed in metacyclics obtained in TAU Pro-Leu, Pro-Ile and Pro-Val when compared to metacyclics obtained in TAU Pro.**

**Down-regulated**

| Gene ID | Product Description | Curated GO Biological Process | Curated GO Cell Component |
|---|---|---|---|
| TcCLB.503959.78 | Hypothetical protein, conserved | N/A | N/A |
| TcCLB.504203.40 | Hypothetical protein, conserved | N/A | GO:0005739 mitochondrion |
| TcCLB.506147.110 | Flagellum attachment zone protein 4 | N/A | GO:0005737; GO:0005856 cytoplasm; cytoskeleton |
| TcCLB.510055.50 | Hypothetical protein, conserved | N/A | GO:0000785 chromatin |
| TcCLB.510687.120 | snoRNP protein GAR1, putative | N/A | GO:0005737; GO:0005634 cytoplasm; nucleus |

**Up-regulated**

| Gene ID | Product Description | Curated GO Biological Process | Curated GO Cell Component |
|---|---|---|---|
| TcCLB.506177.20 | lectin, putative | N/A | N/A |
| TcCLB.508507.40 | Hypothetical protein, conserved | N/A | N/A |
| TcCLB.510101.450 | Hypothetical protein, conserved | N/A | GO:0005930; GO:0000785 axoneme; chromatin |
| TcCLB.510749.30 | Autophagy-related protein 24 | GO:0006914; GO:0006898; GO:0042493 autophagy; receptor-mediated endocytosis; response to drug | GO:0005737; GO:0030666 cytoplasm; endocytic vesicle membrane |

differentiated in TAU Pro-Leu, Pro-Ile and Pro-Val (Table 1, Fig 6). Interestingly, the autophagy-related protein 24 is up-regulated in metacyclics differentiated in the presence of TAU Pro supplemented with any of the BCAAs, whereas the flagellum attachment zone protein 4 (FAZ4) is down-regulated in those parasites (S26 and S27 Tables). Moreover, proline

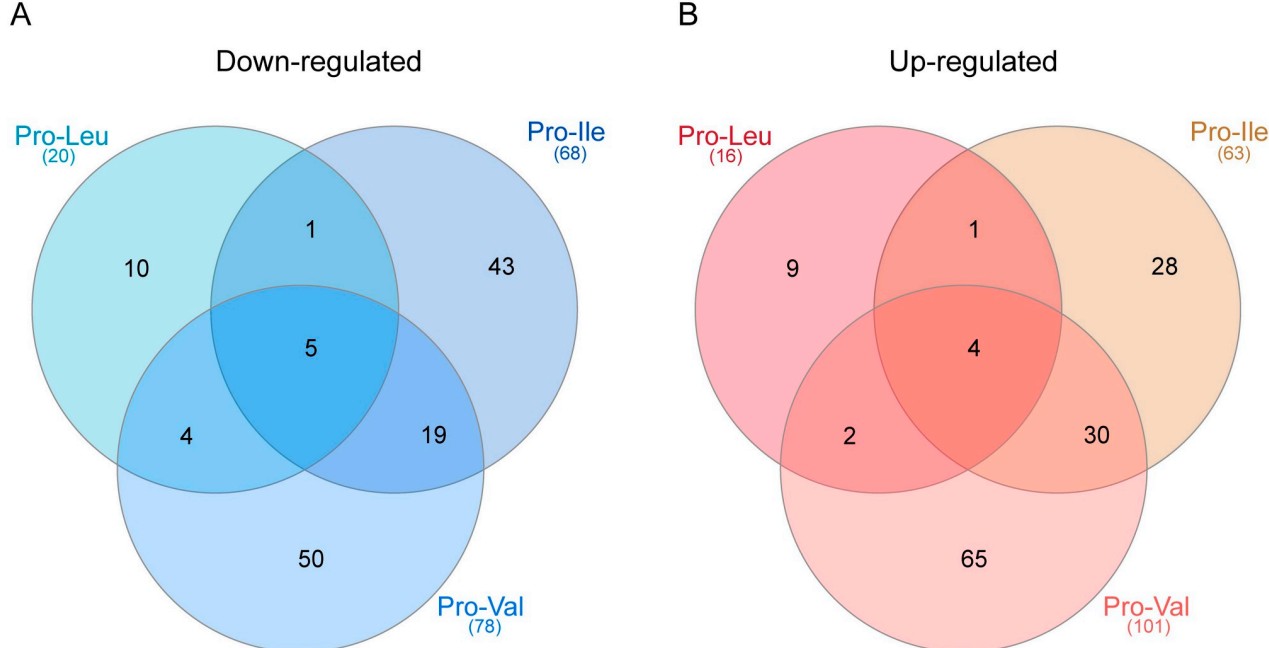

**Fig 6. Each of the BCAAs induces particular proteomic changes in the metacyclic trypomastigotes.** Venn diagrams show the number of proteins shared between the datasets of down (A) and up-regulated (B) proteins of metacyclics differentiated in TAU Pro-Leu, Pro-Ile and Pro-Val when compared to TAU Pro.

dehydrogenase, glutathione-S-transferase and fatty acyl-CoA synthetase are up-regulated in TAU Pro-Ile and TAU Pro-Val metacyclics. Different trans-sialidases are down-regulated in TAU Pro-Leu and TAU Pro-Val when compared with TAU Pro. Furthermore, the enzyme tyrosine aminotransferase (TAT) is up-regulated in parasites differentiated in TAU Pro-Ile and TAU Pro-Val, whereas pyruvate kinase (PK) is down-regulated in parasites differentiated in TAU Pro-Ile when compared with TAU Pro. PK activity and TAT expression were analysed in the crude extract of metacyclics differentiated in TAU Pro and TAU Pro+BCAAs (S4 and S5 Figs). The results confirm the differential expression of PK and TAT identified in the proteomic analysis. Taken together, the data indicate that metacyclogenesis in the presence of each BCAA induces differences in the proteomic profiles of the metacyclic trypomastigotes.

## Discussion

Metacyclogenesis occurs at the final portion of the triatomine's gut. In this adverse environment, the parasite faces intense nutritional stress, which has been shown to trigger cell adhesion and metacyclogenesis [3,33]. Early studies have shown the presence of several amino acids in the excreta of the insect vector *Rhodnius prolixus*, such as histidine, glycine, phenylalanine, leucine and valine [34]. More recently, metabolomic analysis of faeces of different triatomine species showed that phenol lipids and amino acids are the most abundant components of the insect's excreta [35].

Other factors have been shown to impact the metacyclogenesis rate either *in vivo* and/or *in vitro*, such as the trypanosome redox state [36], the temperature in the insect vector [37] and the regulation of autophagy [38,39]. One other important factor which determines the success of cell differentiation is the exposure to different substrates [3,5,40,41]. Some substrates such as the amino acids proline, asparagine and glutamine are 'pro-metacyclogenic', supporting cell differentiation. Other substrates such as the branched-chain amino acids (BCAAs) leucine, isoleucine and valine are 'anti-metacyclogenic' not only because they do not support metacyclogenesis but also inhibit the action of the metabolites that do so. As it has been previously shown, the presence of proline combined with any of the BCAAs during *in vitro* metacyclogenesis reduces cell differentiation rates [4,6]. In this work, we investigated if the metacyclic trypomastigotes originated in these conditions present important differences in their protein level profiles.

Proteomic analysis showed that the proteomic profile between parasites differentiated in TAU 3AAG and TAU Pro medium is very similar, which could be expected since proline is one of the components of TAU 3AAG [28]. However, the comparison between the proteomic profiles of the metacyclics differentiated in TAU Pro versus TAU Pro supplemented with each BCAA presented more than three hundred proteins differentially expressed, suggesting that BCAAs might not only interfere directly in the proline metabolism during metacyclogenesis but also in other biological processes.

Metacyclogenesis induced in TAU Pro supplemented with Leu, Ile or Val resulted in lower differentiation rates when compared to metacyclogenesis induced in TAU Pro. Interestingly, parasites differentiated in TAU Pro and TAU Pro-Leu showed the most similar proteomic profile whilst parasites obtained in the presence of Pro-Val showed the most significantly different proteomic profile. Maybe more strikingly is the fact the different proteins involved in a variety of cell processes are differentially regulated in each specific condition, evidencing that despite the similar phenotype in the reduction of metacyclogenesis induced by proline, each BCAA distinctly affects metacyclogenesis. Some of the biological processes that might be affected will be discussed in more detail because their relevance in the biology of *T. cruzi* has already been demonstrated.

*T. cruzi* uses BCAAs as a carbon and energy source and their catabolism produces intermediates for the tricarboxylic acid cycle [41,42]. The canonic enzyme BCAA transaminase is absent in this parasite; the first step of oxidation of BCAAs is catalyzed by tyrosine aminotransferase (TAT) and aspartate aminotransferase (ASAT) [27,43]. These non-canonical aminotransferases differ in their catalytic competencies towards BCAAs; TAT seems to more readily utilize leucine, while ASAT appears to more actively transaminate isoleucine [27]. Considering that TAT is almost exclusively expressed in epimastigotes and that ASAT is expressed in negligible levels in this stage but that its abundance notably increases in the mammalian stages of this protozoan, the difference in the expression levels and substrate preference might reflect specific nutritional needs throughout the life stages of this pathogen [27]. Once the BCAAs are transaminated, their carbon skeletons undergo different metabolic fates. Theoretically, Leu degradation leads to the formation of the ketogenic metabolites acetoacetate and acetyl-CoA, whilst Ile degradation generates acetyl-CoA and propionyl-CoA which are ketogenic and glucogenic compounds. Finally, Val degradation forms succinyl-CoA which is glucogenic [44]. In trypanosomatids, Leu can be used as a direct precursor and carbon source for sterol and isoprenoid biosynthesis and its chain carbons are also utilized for fatty acid production, whereas Ile is poorly used for sterol synthesis and, when used in an anabolic pathway, is more efficiently channelled to fatty acid biosynthesis [45,46]. In addition, it was reported that Glc causes an increment in the metabolism of Leu to $CO_2$ production, revealing a connection between Leu metabolism and glycolysis [47]. Leu metabolism diverges with respect to Ile and Val by the availability of a specialized acyl-CoA dehydrogenase: isovaleryl-CoA dehydrogenase (IVDH); actually, this fact has been studied in our lab. The differences in the metabolic fates of the BCAAs might help to explain the phenotypic particularities observed in the metacyclic trypomastigotes differentiated in the presence of these amino acids, for example, the reduced infectivity that has been previously observed [6].

Transamination of BCAAs has been shown to play a key role in α-ketoglutarate, glutamate and $NH_3$ homeostasis in different types of cells [48–50]. Interestingly TAT was found to be up-regulated in metacyclic trypomastigotes differentiated in TAU Pro-Ile and TAU Pro-Val, but not in TAU Pro-Leu. Additionally, other enzymes putatively involved in pyruvate and glutamate metabolism are differentially expressed in metacyclics obtained in the presence of the BCAAs, such as the down-regulation of glutamate dehydrogenase (GDH) [51] in TAU Pro-Leu and up-regulation in TAU Pro-Ile, down-regulation of pyruvate kinase 2 [52] in TAU Pro-Ile and up-regulation of glutamine synthetase [53] in TAU Pro-Val. Glutamine synthetase (GS) has been described as an ammonium detoxifier in *T. cruzi* amastigotes [53] so it is reasonable to speculate that GS up-regulation occurs in response to ammonium concentration resulting from the increased amino acid catabolism in parasites differentiated in the presence of Pro+BCAAs.

Atwood et al., identified proteins from different stages of *T. cruzi* by LC-MS/MS and concluded that the differentiation into metacyclic trypomastigotes resulted in an increased abundance of proteins involved in antioxidant defenses [54]. Our results indicate the differential regulation of proteins involved in oxidation-reduction in response to the presence of BCAAs during the differentiation process, confirming the importance of this process for this form of the parasite. Polyamines and thiol-containing molecules have also been shown to be involved in the maintenance of the redox balance of *T. cruzi* [55], which in turn plays a major role in the success of the mammalian host infection [56]. More recently, the metabolism of polyamines has also been implicated as a positive regulator of metacyclogenesis in *T. cruzi* [39]. Our results indicate a differential expression of genes involved in glutathione metabolism, such as glutathione synthetase [57], which is down-regulated in parasites differentiated in the presence of Pro-Ile, whereas a putative glutathione S-transferase is up-regulated in metacyclics

differentiated in TAU Pro-Ile and TAU Pro-Val. The data point out that the presence of BCAAs in the medium led to possible differential regulation of polyamine biosynthesis in metacyclic trypomastigotes.

During nutritional stress and metacyclogenesis, translation is attenuated [58], *T. cruzi* uses proteins previously accumulated in reservosomes as amino acid sources [59] and, accordingly, inhibition of proteasome activity prevents metacyclogenesis [60]. Interestingly, proteins putatively involved in proteolysis such as proteasome activator protein pa26 [61] and the putative proteasome regulatory ATPase subunit 1 are up-regulated in metacyclics differentiated in TAU Pro-Ile and TAU Pro-Val whereas several proteins involved in translation, including ribosomal proteins and eukaryotic initiation factors (eIFs) are down-regulated in these parasites when compared to metacyclics differentiated in TAU Pro. However, this relationship seems to be the opposite in parasites differentiated in TAU Pro-Leu when we compare with metacyclics obtained in TAU Pro and TAU 3AAG (Clusters 2 and 3).

The flagellum of invasive trypomastigotes plays an important role in the infection process, driving the trypanosome to initiate and disseminate the infection in the body [62]. Functional analysis of flagellar proteins has now revealed surprising new roles for the flagellum in trypanosomatids, such as cell morphogenesis, cell division, and immune evasion [63]. Our results show that proteins involved in the flagellum structure are differentially regulated in metacyclic trypomastigotes differentiated in TAU Pro supplemented with Ile or Val. Proteins such as inner arm dynein 5–1, dynein heavy chain and paraxonemal rod are up-regulated in metacyclics differentiated in TAU Pro-Val; only dynein light chain was up-regulated in TAU Pro-Ile derivate trypomastigotes. On the other hand, radial spoke protein 3, C-terminal motor kinesin, outer arm dynein and microtubule-associated protein Gb4 are down-regulated in TAU Pro-Val trypomastigotes and only radial spoke 3 is downregulated in TAU Pro-Ile. Dyneins are described as important for the structure and motility of eukaryotic cilia and flagella [64,65]. The outer and inner arm dyneins bound to the A-tubule of doublet microtubules, walk on the neighbouring B-tubule and provide the driving force for motility [63,66]. Outer arm dyneins are generally uniform in composition and may contain either two or three distinct heavy chains, depending on the organism [67]. In *T. brucei*, the knockdown of the outer dynein arm by RNAi led to the absence of propulsive waves and loss of forward motility [68]. Some assembled protein modules such as radial spokes, nexin links and central pair projections are important for axonemal function through their roles as cross-linkers and regulators of doublet microtubules sliding and bending. [66]. Radial spokes provide a platform for the assembly of signaling proteins and are part of a mechanochemical signal transduction system that regulates dyneins [63,69,70]. Radial spoke 3 has been investigated in *T. brucei* and it is required for parasite motility [68,71]. These proteins contain predicted regulatory domains, supporting the concept that spokes play important roles in signal transduction [71] and are modifiers of dynein activity and flagellar beating [63]. The flagellar attachment zone (FAZ) is an adhesion region of *T. cruzi* epimastigote form where the flagellum emerges from the flagellar pocket and remains attached to the cell body [72]. Studies have suggested that the FAZ plays a role in cellular organization and cytokinesis; FAZ structures are replicated and associated with the new flagellum [73]. Also, flagellum attachment zone protein 4 (FAZ4) is downregulated in all conditions in which the BCAAs were added, which might affect the capability of cell invasion of the parasite, once that is important to parasite motility [74]. The proteins involved in the structure of the flagellum are important for this parasite stage, however, the fact that these proteins are mostly differentially regulated in cells differentiated in TAU Pro-Ile and mostly in TAU Pro-Val needs to be more investigated.

Trans-sialidases are cell surface proteins responsible for the incorporation of sialic acid from host cells into molecules present in the parasite membrane and play a role in immune

evasion and host cell entry mechanisms. Trans-sialidases, TS-like and mucin members are likely to be the most abundant proteins on the surface of *T. cruzi* [75]. This family of proteins display greater variability; proteins of group II that have no trans-sialidase activity are capable of binding to β-galactose, laminin, fibronectin, collagen, cytokeratin and are involved in cell adhesion and invasion [76]. By proteomic analysis, De Godoy et al, report that all of the 18 trans-sialidases identified were up-regulated in metacyclic trypomastigotes when compared with the initial phases of differentiation of *T. cruzi* [77]. Cordero et al reported that in metacyclic forms, 39% of the total expressed proteins are surface proteins (members of the trans-sialidase superfamily, surface glycoproteins, mucin and GP63 protease) confirming that this developmental form expresses a large repertoire of surface glycoproteins involved in the host-cell adhesion and invasion [78]. Additionally, it has been recently shown that knocking out active trans-sialidases impairs the intracellular differentiation from amastigote to trypomastigote [79]. In our case, some specific putative trans-sialidases are downregulated in trypomastigotes obtained in TAU Pro-Leu and TAU Pro-Val when compared to metacyclics obtained in TAU Pro. This might contribute to the phenotype previously observed by our group that metacyclics differentiated in the presence of Pro+BCAAs are less infective [6]. Together, differential regulation of flagellar proteins and trans-sialidases might affect the infection capability of these parasites, but this needs to be further investigated.

When we analysed the global proteomic profiles of the metacyclics differentiated in the presence of the BCAAs, we found that only a few proteins were similarly differentially expressed between the conditions. One of the proteins that is up-regulated in all three conditions is an autophagy-related protein, ATG24. Autophagy has been reported as one of the main processes that regulate metacyclogenesis in *T. cruzi* [38,39] and the ortholog of ATG24 in *T. brucei* has been associated with the inhibition of cell differentiation [80], which indicates that differentiation in the presence of BCAAs might be interfering with the regulation of autophagy.

## Concluding remarks

In this work, we analysed the proteomic profile of trypomastigote metacyclics differentiated in the TAU medium supplemented with different substrates. The result demonstrates that the presence of different metabolites during metacyclogenesis can induce changes in the proteomic profiles of the parasites. Parasites that differentiate in the presence of proline, glutamate, aspartate and glucose (3AAG medium) present a more similar proteomic profile than the parasites that differentiate in proline only, while the parasites differentiate in medium supplemented with proline combined with each BCAA present a different proteomic profile when compared with parasites differentiated in the presence of proline. The presence of proline combined with either leucine, isoleucine or valine in the differentiation medium changes the abundance of proteins related to metabolic processes such as amino acid metabolism, as well as oxidation-reduction and translation, mainly. Taken together these data show that the metabolites present in the differentiation medium can modify protein level profiles which can determine the differentiation rates and might affect the infectivity of the parasites.

## Financial disclosure

This work was supported by: Fundação de Amparo à Pesquisa do Estado de São Paulo (FAPESP) 2021/12938-0 (awarded to AMS), and 2018/15553-9 (awarded to JCPC) Conselho Nacional de Pesquisas Científicas e Tecnológicas (CNPq) grant 307487/2021-0 (awarded to AMS) and Wellcome Trust grant 222986/Z/21/Z (awarded to JFN and AMS). The funders had no role in study design, data collection and analysis, decision to publish, or preparation of the

manuscript. JFN is a Wellcome Trust fellow. FSD, SM and FNLV were FAPESP fellows during the development of this work.

## Supporting information

**S1 Fig. Presence of BCAAs affect metacyclogenesis induced in TAU Pro.** Metacyclogenesis efficiency of parasites differentiated in TAU 3AAG, Pro, Pro-Leu, Pro-Ile and Pro-Val. Graph shows average and standard deviation of three biological replicates. Statistically analysis using differentiation rate in TAU Pro as control was performed applying one-way ANOVA with Turkey's multiple comparisons test (a = 0.05).
(PDF)

**S2 Fig. TAU-3AAG and TAU-Pro induce similar proteomic profiles in metacyclic trypomastigotes.** Volcano plot of LFQ values of proteins from metacyclic trypomastigotes differentiated in TAU Pro and TAU 3AAG. Differentially expressed proteins are shown in orange (down-regulated in TAU Pro) and green dots (up-regulated in TAU Pro). Pie charts show the number of hypothetical proteins in the sets of up and down-regulated proteins.
(PDF)

**S3 Fig. Heatmaps of A. GO Biological Process and B. GO Cellular Component.** The frequency of each GO was normalized over the number of differentially expressed proteins in each group (Pro-Leu, Pro-Ile and Pro-Val).
(PDF)

**S4 Fig. Biological validation of TAT protein levels in total extract of metacyclics differentiated in the presence of TAU Pro-BCAAs. A.** Representative image of western blot for tyrosine aminotransferase (TAT) (~47 KDa) with total protein loading control stained with trichloroethanol (TCE) (left) and relative expression of TAT in the total extract of two biological replicates of metacyclics parasites (right). **B.** Average LFQ values for the four annotated TATs in metacyclics differentiated in TAU Pro and TAU Pro-BCAAs. Graphs show average and standard deviation of three biological replicates. Statistically analysis using TAU Pro as the control was performed applying one-way ANOVA with multiple comparisons test (a = 0.05, ** $p \leq 0.01$, *** $p \leq 0.001$).
(PDF)

**S5 Fig. Biological validation of PK protein levels in total extract of metacyclics differentiated in the presence of TAU Pro-BCAAs. A.** Pyruvate kinase (PK) activity in total protein extracts of metacyclics differentiated in TAU Pro and TAU Pro-BCAAs. Graphs show average and standard deviation of two biological replicates, with two technical replicates each. Statistically analysis using TAU Pro as the control was performed applying Turkey's multiple comparisons test (a = 0.05, * $p \leq 0.05$). **B.** Average LFQ values for the two annotated PK paralogues in metacyclics differentiated in TAU Pro and TAU Pro-BCAAs. Graphs show average and standard deviation of three biological replicates. Statistically analysis using TAU Pro as the control was performed applying one-way ANOVA with multiple comparisons test (a = 0.05, ** $p \leq 0.01$, *** $p \leq 0.001$).
(PDF)

**S1 Table. Protein Groups Matrix.**
(XLSX)

**S2 Table. Fuzzy c-mean clustering.**
(XLSX)

**S3 Table. Cluster 1—Gene Ontology.**
(XLSX)

**S4 Table. Cluster 2—Gene Ontology.**
(XLSX)

**S5 Table. Cluster 3—Gene Ontology.**
(XLSX)

**S6 Table. Cluster 4—Gene Ontology.**
(XLSX)

**S7 Table. Cluster 5—Gene Ontology.**
(XLSX)

**S8 Table. Cluster 6—Gene Ontology.**
(XLSX)

**S9 Table. Cluster 7—Gene Ontology.**
(XLSX)

**S10 Table. Downregulated proteins in TAU Pro x TAU 3AAG.**
(XLSX)

**S11 Table. Upregulated proteins in TAU Pro x TAU 3AAG.**
(XLSX)

**S12 Table. Down-regulated proteins in TAU Pro x TAU 3AAG—Gene Ontology.**
(XLSX)

**S13 Table. Up-regulated proteins in TAU Pro x TAU 3AAG—Gene Ontology.**
(XLSX)

**S14 Table. Downregulated proteins in TAU Pro-Leu.**
(XLSX)

**S15 Table. Upregulated proteins in TAU Pro-Leu.**
(XLSX)

**S16 Table. Down-regulated proteins in TAU Pro-Leu—Gene Ontology.**
(XLSX)

**S17 Table. Up-regulated proteins in TAU Pro-Leu—Gene Ontology.**
(XLSX)

**S18 Table. Downregulated proteins in TAU Pro-Ile.**
(XLSX)

**S19 Table. Upregulated proteins in TAU Pro-Ile.**
(XLSX)

**S20 Table. Down-regulated proteins in TAU Pro-Ile—Gene Ontology.**
(XLSX)

**S21 Table. Up-regulated proteins in TAU Pro-Ile—Gene Ontology.**
(XLSX)

**S22 Table. Downregulated proteins in TAU Pro-Val.**
(XLSX)

**S23 Table. Upregulated proteins in TAU Pro-Val.**
(XLSX)

**S24 Table. Down-regulated proteins in TAU Pro-Val—Gene Ontology.**
(XLSX)

**S25 Table. Up-regulated proteins in TAU Pro-Val—Gene Ontology.**
(XLSX)

**S26 Table. Venn diagram—Down-regulated proteins.**
(XLSX)

**S27 Table. Venn diagram—Up-regulated proteins.**
(XLSX)

## Acknowledgments

This work would not have been possible without the invaluable resources of VEuPathDB, especially the TriTrypDB. These databases have been supporting the advances in the parasitology field and are essential for the progress of the research on Neglected Tropical Diseases.

## Author Contributions

**Conceptualization:** Janaina de Freitas Nascimento, Ariel Mariano Silber.

**Data curation:** Janaina de Freitas Nascimento, Flávia Silva Damasceno, Renan Weege Achjian, Julia Pinheiro Chagas da Cunha.

**Formal analysis:** Janaina de Freitas Nascimento, Flávia Silva Damasceno, Sabrina Marsiccobetre, Francisca Natália de Luna Vitorino, Renan Weege Achjian, Julia Pinheiro Chagas da Cunha, Ariel Mariano Silber.

**Funding acquisition:** Janaina de Freitas Nascimento, Julia Pinheiro Chagas da Cunha, Ariel Mariano Silber.

**Investigation:** Janaina de Freitas Nascimento, Flávia Silva Damasceno, Sabrina Marsiccobetre, Francisca Natália de Luna Vitorino, Julia Pinheiro Chagas da Cunha, Ariel Mariano Silber.

**Methodology:** Janaina de Freitas Nascimento, Flávia Silva Damasceno, Francisca Natália de Luna Vitorino, Renan Weege Achjian.

**Project administration:** Janaina de Freitas Nascimento, Ariel Mariano Silber.

**Resources:** Janaina de Freitas Nascimento, Julia Pinheiro Chagas da Cunha, Ariel Mariano Silber.

**Software:** Janaina de Freitas Nascimento, Renan Weege Achjian, Julia Pinheiro Chagas da Cunha, Ariel Mariano Silber.

**Supervision:** Janaina de Freitas Nascimento, Julia Pinheiro Chagas da Cunha, Ariel Mariano Silber.

**Validation:** Janaina de Freitas Nascimento, Flávia Silva Damasceno, Sabrina Marsiccobetre, Francisca Natália de Luna Vitorino, Renan Weege Achjian, Julia Pinheiro Chagas da Cunha, Ariel Mariano Silber.

**Visualization:** Janaina de Freitas Nascimento, Ariel Mariano Silber.

**Writing – original draft:** Janaina de Freitas Nascimento, Flávia Silva Damasceno, Sabrina Marsiccobetre, Julia Pinheiro Chagas da Cunha, Ariel Mariano Silber.

**Writing – review & editing:** Janaina de Freitas Nascimento, Ariel Mariano Silber.

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
