## [Decision Letter · Decision Letter 0]

20 Sep 2024

Dear Prof. Silber,

Thank you very much for submitting your manuscript "Branched chain amino acids modulate the proteomic profile of Pro-induced differentiation for the infective stages of Trypanosoma cruzi" for consideration at PLOS Neglected Tropical Diseases. As with all papers reviewed by the journal, your manuscript was reviewed by members of the editorial board and by several independent reviewers. The reviewers appreciated the attention to an important topic. Based on the reviews, we are likely to accept this manuscript for publication, providing that you modify the manuscript according to the review recommendations. 

Sincerely,

Álvaro Acosta-Serrano

Section Editor

Reviewer's Responses to Questions

**Key Review Criteria Required for Acceptance?**

**Methods**

-Are the objectives of the study clearly articulated with a clear testable hypothesis stated?

-Is the study design appropriate to address the stated objectives?

-Is the population clearly described and appropriate for the hypothesis being tested?

-Is the sample size sufficient to ensure adequate power to address the hypothesis being tested?

-Were correct statistical analysis used to support conclusions?

-Are there concerns about ethical or regulatory requirements being met?

Reviewer #1: Methods all fine.

Reviewer #2: The experimental workflow is adequate, the data analysis robust and comparisons revealed a proteome remodelling snapshot where valine emerged as the most influential treatment. Overall protein coverage was 7.6% (1,457 out of 19,242 proteins for CL-Brener proteome). Changes in protein abundances depict a phenotype mainly driven by metabolic responses, attenuated translation and proteolytic rates, and changes in flagellar and parasite cell surface components. While the study didn't pinpoint exactly which proteins control the parasites' transformation, it still provides valuable insights. On a technical note, trypanosomes proteome is poorly annotated, and this type of studies may improve the limited annotation present in nearly 33% of hypothetical proteins.

**Results**

-Does the analysis presented match the analysis plan?

-Are the results clearly and completely presented?

-Are the figures (Tables, Images) of sufficient quality for clarity?

Reviewer #1: In part. As detailed in my comments ot authors, the basic analysis of fine, it it the higher level interoretation that is problematic.

Reviewer #2: (No Response)

**Conclusions**

-Are the conclusions supported by the data presented?

-Are the limitations of analysis clearly described?

-Do the authors discuss how these data can be helpful to advance our understanding of the topic under study?

-Is public health relevance addressed?

Reviewer #1: The conclusions are not fully realised or connected with the data.

Reviewer #2: (No Response)

**Editorial and Data Presentation Modifications?**

Reviewer #1: (No Response)

Reviewer #2: The data presented warrants publication in PNTD and I have made a few suggestions that can aid readers to better grasp the most relevant outputs of this study.

Major points:

- The proteomic data provide evidence of a potential role for TAT as a frontline sensing component when free amino acids are the main carbon and nitrogen sources found in the culture media. It will be informative to presenting an average LFQ comparison across the media conditions assayed for TAT and other metabolic components such as PK which stood out as proteins whose abundance was remarkably changed. Bars plot would be enough.

- The FAZ4, Gb4 and autophagy-related protein 24 are additional candidates that can be featured from this study in both abstract and the discussion sections. Is there any common feature in domain architecture or subcellular localisation among these? These protein hits might be worth studying further.

- It is fascinating to see that a single isomeric modification (i.e., leucine x isoleucine) can be decoded by the parasite and finely influence proteome changes. Another highlight.

Minor points:

- The use of the acronym ‘pro’ is excessive and might add some confusion as this can also be interpreted as synonym of enhancer/promoter/inducer. I suggest naming it on its full extension throughout the main sections (tittle, abstract, conclusions). For example, sentence in line 106: ‘Given that i) Pro is a pro-metacyclogenic amino acid’ can be misleading.

- Using the term ‘protein expression’ is not adequate for the context of this type of study. Although it’s been widely used the word expression should only refer to gene expression. In the context of gene-encoded proteins I would suggest replacing expression for protein synthesis, protein levels or protein abundance. Protein expression is conceptually flawed. 

- Line 340: How do the authors infer metabolic changes from proteome data? The variations observed in enzymes of glutamate, proline, serine, threonine and glutamine metabolism reflect an overall metabolic remodelling that, in my view, is mostly driven by the metabolic condition imparted in the cultivation media. 

- Comparisons between protein changes observed in TAU-proline x TAU-proline+valine (Lines 360-70) are not properly presented as the changes highlighted do not specify in what condition the up/down regulated hits are enriched. For example, in which comparison were the 78 downregulated hits seen? Similarly written in line 376. Please note that data presented in lines 389-409 are much easier to follow on and correlate with the supporting figures. 

- Line 459-60 (Discussion section) where the authors state the difference in substrate preferences might reflect specific nutritional needs throughout life cycle. I think it would be useful to elaborate more around this claim and, if possible, provide a documented example. 

- Do the authors think that an increased amino acid catabolic activity can be a trigger of metacyclogenesis arrest? This is put in the context of increased GS activity in the TAU-proline plus valine dependent metacyclogenesis.

**Summary and General Comments**

Reviewer #1: This is an interesting paper that seeks to address the question of the impact of different amino acids on differentiation in T. cruzi. Given that differentiation is frequently associated with stress conditions, uderstanding the metabolic consequences required for and arising from differentiation is important to understand the basic biology behind the process. Technically I have no real criticisms or comments as the proteomics approach is absolutely standard and clearly carried out correctly. There is some noise revealed in the PCA analysis in F1 revealing an outlier for several of the conditions, but that is unlikely to strongly impact the data. 

My real issue is to do with the manner in which the data are considered. As we work through the results there is little interpretation, but when we come to the discussion I was struck that the vast majorty of this coulle have been written without the proteomics having been done. There is little referemce back to this except for the section on the flagellar changes. I also worry that considerable issues arise from the use of GO terms which are poor for kinetoplastids and a lack of detailed analysis of more prominant changes. This seems to be a missed opportunity and I would suggest improving the discussion; the authors are well respected in this area and their insights would be valuable. 

Some specific things are below;

Suggest Pro-induced is a bit cryptic in the title. 

Grammer odd for first sentence of abstract. 

down-regulated proteins are involved in biological processes such as the generation of energy, posttranscriptional regulation of gene expression and oxidation-reduction processes; on the other hand, up-regulated proteins participate in respiration and translation processes

Not sure I quite grasp the distinction here - respiration is an energy genearator and posttranscriptional events include translation. 

Define TAT in discussion. 

The differences in the metabolic fates of the BCAAs might explain the phenotypic particularities observed in the metacyclic trypomastigotes differentiated in the presence of these amino acids.

Yes, but how?

In our case, some specific putative trans-sialidases proteins are downregulated in trypomastigotes obtained in TAU Pro-Leu and TAU Pro-Val when compared to metacyclics obtained in TAU Pro

How does the present dataset compare to the other studies cited?

Odd that if autophagy is activated that only see ATG24 and none of the catabolic or trafficking enzymes are noted.

Reviewer #2: Professor Silber’s research group has greatly increased our understanding of how amino acids influence the trypanosomatids’ life cycle. Multiple lines of evidence have demonstrated that its functionality goes beyond its proteogenic role playing a pivotal role in parasite proliferation/infectivity, energy production, osmoregulation, and stress response. Particularly, proline acts as modulator of parasite differentiation and mitochondrial homeostasis.

In this study the authors applied bottom-up proteomic profiling to delve into the function of branched chain amino acids (leucine, isoleucine and valine) during the differentiation process known as metacyclogenesis (i.e., cellular transformation from epimastigote to metacyclic trypomastigote). For this characterisation authors applied a well-established system of axenic cultivation in nutrient-scarce media, followed by affinity-based cell purification, which enabled the assessment of proteome-wide changes upon proline-dependent metacyclogenesis.

PLOS authors have the option to publish the peer review history of their article (what does this mean?). If published, this will include your full peer review and any attached files.

Reviewer #1: No

Reviewer #2: No

Figure Files:

Data Requirements:

Reproducibility:

References

---

## [Editor Report · Decision Letter 1]

30 Sep 2024

Dear Prof. Silber,

We are pleased to inform you that your manuscript 'Branched-chain amino acids modulate the proteomic profile of Trypanosoma cruzi metacyclogenesis induced by proline' has been provisionally accepted for publication in PLOS Neglected Tropical Diseases.

Best regards,

Álvaro Acosta-Serrano

Section Editor

---

## [Editor Report · Acceptance letter]

5 Oct 2024

Dear Prof. Silber,

We are delighted to inform you that your manuscript, "Branched-chain amino acids modulate the proteomic profile of Trypanosoma cruzi metacyclogenesis induced by proline," has been formally accepted for publication in PLOS Neglected Tropical Diseases.

Best regards,

Shaden Kamhawi

co-Editor-in-Chief

Paul Brindley

co-Editor-in-Chief
